# OPENFOLEY: OPEN-SET VIDEO-TO-AUDIO GENERATION WITH MODALITY-AWARE MASKING AND FLOWS

## ABSTRACT

Video-to-audio generation has emerged as a promising frontier for enriching multi-modal understanding and synthesis. However, most existing approaches operate under closed-set assumptions, restricting training and evaluation to predefined categories and limiting generalization in open-world scenarios. Prior methods primarily rely on pre-trained vision-language or audio-language encoders such as CLIP and CLAP, overlooking the strong inherent video–audio correspondence that can directly guide cross-modal grounding. In this work, we present *OPENFOLEY*, a novel framework for open-set video-to-audio generation that enforces semantic fidelity and rhythmic synchronization across modalities. Our approach introduces a modality-aware dynamic masking strategy, where audio segments are reconstructed from masked video frames and vice versa, enabling the model to capture fine-grained temporal alignment without relying solely on external encoders. Furthermore, we design a generalized masked flow-based module that conditions generation on selectively sampled video frames, significantly improving efficiency and fidelity while preserving cross-modal coherence. Comprehensive experiments on VGGSound and a newly curated open-set benchmark demonstrate that *OPENFOLEY* consistently outperforms state-of-the-art baselines in both objective and perceptual metrics, achieving superior Fréchet Audio Distance (FAD) and Kullback–Leibler (KL) divergence scores. The project page can be found at: https://openfoley.github.io.

## 1 INTRODUCTION

The ability to generate realistic audio from visual input is a fundamental challenge in multimodal learning, with broad applications spanning virtual reality, accessibility, immersive entertainment, and creative media production. Video-to-audio generation requires not only capturing semantic cues from visual scenes but also synthesizing temporally aligned and perceptually plausible sounds. While recent years have seen encouraging progress, most existing approaches operate under a *closed-set* assumption, where training and evaluation are restricted to predefined categories. This setup limits their capacity to generalize to the diverse and unpredictable nature of real-world scenarios. In practice, effective video-to-audio generation demands *open-set generalization*: the ability to synthesize synchronized and semantically consistent audio for previously unseen visual content.

Early advances in video-to-audio generation have largely relied on cross-modal encoders such as CLIP (Radford et al., 2021) and CLAP (Wu et al., 2023), which align vision-language and audio-language modalities, respectively. These encoders provide strong priors for semantic reasoning, enabling models to capture high-level correspondences between what is seen and what is heard (Iashin & Rahtu, 2021; Sheffer & Adi, 2023; Kreuk et al., 2023; Luo et al., 2023). However, their reliance on language as an intermediary introduces fundamental limitations. Since CLIP and CLAP are optimized for text-mediated associations, they fail to capture the intrinsic video–audio correspondence that arises naturally from co-occurring motion and sound. As a result, current models often produce audio that is semantically plausible but poorly synchronized with fine-grained visual dynamics, such as footsteps lagging behind walking motions or mismatched timing between object interactions and sound effects. Furthermore, language-based encoders tend to emphasize global semantics, overlooking the rhythmic and temporal patterns essential for perceptually convincing video-to-audio generation.

The challenges of *open-set* video-to-audio generation exacerbate these limitations. In contrast to closed-set training, where category-specific priors can be memorized, open-set scenarios require models to adapt to unseen scenes, objects, and motion patterns without explicit supervision. Achieving this goal entails learning robust cross-modal representations that remain generalizable while preserving temporal synchronization. Moreover, unseen content may introduce novel environmental contexts, background sounds, and interaction dynamics that demand flexible modeling beyond category-dependent alignment. Existing methods, with their reliance on indirect language-based mappings and category-constrained training, fall short of addressing these requirements.

To overcome these challenges, we propose OPENFOLEY, a novel framework for open-set video-to-audio generation that directly enforces both semantic coherence and temporal synchronization between modalities. Our key innovation is a *modality-aware dynamic masking strategy*, in which masked audio segments are reconstructed from video frames, and masked video features are predicted from audio. This bidirectional masking encourages the model to discover fine-grained temporal correspondences by leveraging the natural co-occurrence of visual and auditory cues, rather than relying solely on language supervision. In addition, we introduce a *generalized masked flow-based module*, which conditions audio generation on selectively sampled video frames. This flow-based conditioning not only improves generation efficiency and fidelity but also enhances the model's ability to remain synchronized with the video stream across diverse and unseen scenarios.

We validate OPENFOLEY on the widely used VGGSound dataset and introduce a new open-set benchmark specifically designed to evaluate generalization beyond predefined categories. Experimental results demonstrate that OPENFOLEY achieves state-of-the-art performance, significantly improving metrics such as Fréchet Audio Distance (FAD) and Kullback–Leibler (KL) divergence compared to existing baselines. Qualitative analyses further confirm that OPENFOLEY generates audio that is both semantically consistent and temporally aligned, even when faced with novel visual content. By highlighting the importance of direct video–audio alignment and robust multimodal representation learning, our work establishes a new direction for open-set video-to-audio generation and lays the groundwork for future research in this emerging domain.

Overall, we summarize our contributions below:

- We introduce OPENFOLEY, a novel framework for *open-set video-to-audio generation* that directly enforces semantic coherence and temporal synchronization between modalities, moving beyond prior methods that rely solely on language-mediated alignment.
- We propose two key components: (i) *Modality-aware Masking Alignment (MMA)*, which enforces cross-modal reconstruction and captures fine-grained synchronization patterns, and (ii) *Modality-aware Flow Generation (MFG)*, which provides a flexible prior for efficient and high-fidelity audio synthesis.
- We curate a new open-set benchmark from AudioSet and Panda70M, and demonstrate that OPENFOLEY achieves state-of-the-art performance across KLD, FAD, and alignment accuracy, with ablations confirming the complementary benefits of MMA and MFG.

## 2 RELATED WORK

**Video-to-Audio Generation.** Video-to-audio generation, the task of translating visual information into corresponding audio outputs, has advanced rapidly with the rise of generative modeling. SpecVQ-GAN (Iashin & Rahtu, 2021) employs vector quantized GANs to transform visual features into audio spectrograms, while Im2Wav (Sheffer & Adi, 2023) leverages CLIP embeddings to directly generate audio waveforms from images. Diff-Foley (Luo et al., 2023) combines Contrastive Audio-Visual Pre-training (CAVP) with diffusion models to refine synchronization between modalities. Foley-Gen (Mei et al., 2023) uses a neural audio codec for waveform-to-token conversion and models audio generation as a language modeling problem. Recent work has shifted toward multimodal pre-training and cross-modal alignment. Seeing & Hearing (Xing et al., 2024) employs the pre-trained ImageBind model (Girdhar et al., 2023) as a latent aligner for cross-modal diffusion-based generation. VAB (Su et al., 2024) introduces masked audio token prediction conditioned on visual features for pre-training without diffusion. MaskVAT (Pascual et al., 2024) combines a sequence-to-sequence masked generative model with a neural codec to improve temporal synchronicity. VATT (Liu et al., 2024) integrates large language models such as Gemma-2B (Team et al., 2024) and LLaMA-2-7B (Touvron et al., 2023) with projection layers to map video features into audio tokens.

While these methods demonstrate impressive progress, they primarily operate under closed-set assumptions and rely on language-mediated embeddings (CLIP/CLAP) for alignment. As a result, they capture global semantics but struggle with fine-grained synchronization and fail to generalize to unseen categories. In contrast, OPENFOLEY introduces *modality-aware masking alignment* to directly enforce cross-modal reconstruction and *modality-aware flow generation* to learn a flexible audio prior. Together, these components enable robust temporal alignment and perceptual quality in both closed- and open-set video-to-audio generation.

**Diffusion Models.** Diffusion models have emerged as a powerful paradigm for generative modeling across multiple domains. The foundational denoising diffusion probabilistic models (DDPMs) (Ho et al., 2020; Song et al., 2021) introduced a forward–reverse process of noise corruption and denoising, and have since powered breakthroughs in image synthesis (Saharia et al., 2022), image restoration (Saharia et al., 2021), speech synthesis (Kong et al., 2021), and video generation (Ho et al., 2022). Building on these foundations, diffusion models have been extended to cross-modal generation tasks. AudioGen (Kreuk et al., 2023) explores diffusion for text-to-audio generation, while DiffSound (Yang et al., 2022) applies diffusion to conditional sound synthesis. In the video–audio domain, Diff-Foley (Luo et al., 2023) integrates contrastive audio-visual pre-training with diffusion decoding to generate synchronized audio from video.

Our work builds on this line of research but diverges in two crucial ways. First, rather than relying exclusively on diffusion with Gaussian priors, we incorporate *flow-based generation* to learn a more expressive and adaptive latent distribution, improving perceptual realism and efficiency. Second, we pair this with *modality-aware masking*, which enforces bidirectional reconstruction across video and audio, ensuring synchronization even in open-set conditions. By combining masking-based alignment with flow-based generation, OPENFOLEY establishes a new paradigm for generalizable synthesis.

# 3 METHOD

In this section, we introduce OPENFOLEY, a novel framework for open-set video-to-audio generation that enforces both semantic coherence and temporal synchronization through modality-aware masking alignment and flow-based audio generation. We first provide preliminaries in Section 3.1, then present Modality-aware Masking Alignment in Section 3.2 to learn semantic and rhythmic coherence, and finally introduce Modality-aware Flow Generation in Section 3.3 to accelerate the video-to-audio generation process while preserving fidelity.

## 3.1 PRELIMINARIES

In this section, we first describe the problem setup and notations, and then revisit the video/audio masked pre-training and the flow-based denoising diffusion probabilistic models.

**Problem Setup and Notations.** Let $\mathcal{V} = \{v_t\}_{t=1}^T$ denote a sequence of video frames, where $v_t \in \mathbb{R}^{H \times W \times C}$ represents the $t$-th frame with height $H$, width $W$, and $C$ color channels. Similarly, let $\mathcal{A} = \{a_t\}_{t=1}^T$ denote the corresponding audio waveform segments. Given an input video sequence $\mathcal{V}$, our goal is to generate a plausible audio sequence $\hat{\mathcal{A}}$ that aligns semantically and temporally with the visual content. We model this generation as a conditional probabilistic process: $P(\hat{\mathcal{A}}|\mathcal{V}; \theta)$, where $\theta$ represents the learnable parameters of the model. Unlike previous works that rely on pre-trained vision-language and audio-language encoders, our method directly learns video-audio alignment through structured masking and flow-based modeling.

**Flow-Based Diffusion Model.** Diffusion probabilistic models (DPMs) (Ho et al., 2020) have achieved decent performance in generative modeling, particularly for speech and audio synthesis. In a standard diffusion model, the forward process gradually adds Gaussian noise to an audio signal $\mathcal{A}$ over $T$ timesteps:

$$q(a_t|a_{t-1}) = \mathcal{N}(a_t; \sqrt{\alpha_t}a_{t-1}, (1-\alpha_t)I), \tag{1}$$

where $\alpha_t$ controls the noise schedule. The goal of the reverse process is to denoise the signal step by step to recover the original data:

$$p_\theta(a_{t-1}|a_t) = \mathcal{N}(a_{t-1}; \mu_\theta(a_t, t), \Sigma_\theta(a_t, t)). \tag{2}$$

However, diffusion models suffer from slow inference speeds, as generating a single sample requires iterating through hundreds or thousands of denoising steps. To accelerate this, a flow-based model has been explored as an alternative. Normalizing flows learn an invertible transformation between the data space and a simpler latent distribution:

$$z = f_\theta(\mathcal{A}), \quad \mathcal{A} = f_\theta^{-1}(z), \tag{3}$$

where $z$ follows a known distribution (*e.g.*, Gaussian). This allows for fast sampling and efficient representation learning. Hybrid approaches that combine diffusion models with flow-based priors have shown promise in reducing the number of required denoising steps while preserving high-fidelity synthesis. Despite these advances, existing audio diffusion models lack explicit temporal synchronization mechanisms with video inputs. Most prior work conditions audio synthesis on category labels or text descriptions, which do not capture the fine-grained motion and rhythmic cues present in videos. This motivates the need for a modality-aware approach that can enforce direct video-audio alignment.

## 3.2 Modality-aware Masking Alignment

Inspired by masked modeling objectives in self-supervised learning, we introduce a **modality-aware dynamic masking** strategy to jointly learn semantic coherence and rhythmic synchronization between video and audio. Unlike traditional masked modeling that reconstructs inputs within the same modality, our approach enforces *cross-modal reconstruction*, where the masked segments in one modality must be predicted from the other. This design ensures that the model captures intrinsic video–audio correspondences rather than relying on indirect textual embeddings or category priors.

Formally, given a masked video sequence $\tilde{\mathcal{V}}$ and its corresponding masked audio sequence $\tilde{\mathcal{A}}$, the model is trained to reconstruct the missing content using cross-modal cues:

$$\mathcal{L}_{\text{mask}} = \mathbb{E}_{\mathcal{V},\mathcal{A}} \left[ \|\hat{\mathcal{A}} - \mathcal{A}\|_2^2 + \|\hat{\mathcal{V}} - \mathcal{V}\|_2^2 \right], \tag{4}$$

where $\hat{\mathcal{A}}$ and $\hat{\mathcal{V}}$ are the reconstructed audio and video features, respectively. This objective aligns the learning signal across modalities, encouraging the network to capture synchronization patterns that emerge from natural co-occurrence.

**Bidirectional Masking.** To explicitly enforce video–audio alignment, we adopt a bidirectional masking strategy:

1. Audio Reconstruction from Video: A random subset of the audio sequence $\mathcal{A}$ is masked (indices $\mathcal{M}_a$), and the model predicts the missing waveforms or spectral features conditioned on visual context $\mathcal{V}$.

2. Video Prediction from Audio: Conversely, a random subset of video frames $\mathcal{V}$ is masked (indices $\mathcal{M}_v$), and the model predicts the missing visual features using surrounding frames and unmasked audio $\mathcal{A}$.

The corresponding loss is:

$$\mathcal{L}_{\text{ma-mask}} = \mathbb{E}_{\mathcal{V},\mathcal{A}} \left[ \sum_{t \in \mathcal{M}_a} \|\hat{a}_t - a_t\|_2^2 + \sum_{t \in \mathcal{M}_v} \|\hat{v}_t - v_t\|_2^2 \right], \tag{5}$$

where $\mathcal{M}_a$ and $\mathcal{M}_v$ denote the sets of masked audio and video indices, respectively.

**Dynamic and Modality-Specific Masking.** Instead of masking uniformly, OpenFoley employs *dynamic masking* tailored to each modality:

- For audio, we mask contiguous segments in the time domain to simulate dropouts in rhythmic or environmental cues, forcing the model to infer missing sound dynamics from video motion.

- For video, we mask both random individual frames and temporally contiguous clips, ensuring the model learns to interpolate motion from auditory patterns and neighboring frames.

This dynamic masking reflects real-world uncertainty where visual or auditory information may be partially occluded or absent. By making the masking modality-aware, we ensure that the learned cross-modal representations are robust and adaptable.

**Benefits for Open-Set Generalization.** Unlike category-dependent supervision, modality-aware masking teaches the model to *reason across modalities* rather than memorize closed-set categories. This makes OpenFoley inherently more capable of handling novel or unseen categories: the model relies on learned synchronization cues (*e.g.*, footsteps matching walking motion, collision sounds following impact) instead of fixed semantic labels. As a result, modality-aware masking acts as a strong inductive bias for open-set video-to-audio generation, improving temporal coherence, semantic fidelity, and robustness to unseen content.

### 3.3 MODALITY-AWARE FLOW GENERATION

While modality-aware masking alignment enforces cross-modal grounding, it does not directly address the challenge of generating high-fidelity audio efficiently. Diffusion-based methods often require hundreds of denoising steps and rely on Gaussian priors that do not capture the rich variability of natural sounds. To improve both efficiency and quality, we integrate a flow-based generative module that reshapes the audio latent space into a more structured distribution, enabling more accurate sampling and faster convergence.

**Normalizing Flow for Audio Latents.** Instead of assuming a standard Gaussian prior, we employ an invertible transformation $f_\theta$ to map audio features $\mathcal{A}$ into a tractable latent distribution:

$$z = f_\theta(\mathcal{A}), \quad \mathcal{A} = f_\theta^{-1}(z), \tag{6}$$

where $f_\theta$ is implemented as a stack of bijective flow layers (e.g., affine coupling or continuous-time flows). This formulation allows exact likelihood estimation and enables us to learn a data-adaptive prior that captures the multimodal nature of real-world audio. In practice, this improves the expressiveness of the generative model, especially in open-set scenarios where unseen audio categories may not conform to Gaussian assumptions.

**Selective Frame Conditioning.** To reduce the complexity of full-sequence modeling, we introduce a modality-aware conditioning mechanism that leverages only *key video frames*. Instead of conditioning on all frames, which is computationally redundant and may introduce noise, our method identifies salient frames that correspond to strong motion or scene transitions. This selective conditioning reduces redundancy while ensuring that temporal anchors (e.g., object collisions, footsteps, or speech articulation) are preserved.

**Iterative Flow Refinement.** Given an initial latent $z_0$ sampled from the flow prior, audio generation is refined iteratively by incorporating video-conditioned updates:

$$z_{t+1} = z_t + \lambda \cdot g_\theta(v_t), \tag{7}$$

where $g_\theta$ is a learned transformation conditioned on the selected video frame $v_t$, and $\lambda$ is a scaling factor that controls the strength of visual guidance. This iterative refinement can be seen as a residual correction process that progressively aligns audio dynamics with visual cues. Unlike global conditioning approaches, this localized refinement explicitly synchronizes transient events in the audio with corresponding visual signals.

**Training Objective.** The flow-based module is trained jointly with the masking alignment strategy by minimizing a conditional likelihood objective:

$$\mathcal{L}_{\text{flow}} = \mathbb{E}_{\mathcal{V},\mathcal{A}} \left[ -\log p_\theta(f_\theta(\mathcal{A}) \mid \mathcal{V}) \right], \tag{8}$$

which encourages the flow to model the distribution of audio latents conditioned on video context. This complements the reconstruction-based losses in Section 3.2, yielding both robust cross-modal representations and efficient generation.

By learning a structured latent distribution via flows, the model avoids over-reliance on closed-set category priors and instead focuses on flexible mappings between visual and auditory domains. Selective frame conditioning ensures synchronization with visual dynamics, while iterative refinement reduces sampling overhead. Together, these properties make the flow-based module especially effective for *open-set video-to-audio generation*, where efficiency, fidelity, and generalization to unseen scenarios are equally critical.

Table 1: Comparison results on VGGSound test set for video-to-audio generation. The KLD, FAD, and Align Acc values are reported.

| Method | KLD ↓ | FAD ↓ | Align Acc ↑ |
|---|---|---|---|
| SpecVQGAN (Iashin & Rahtu, 2021) | 3.78 | 6.63 | 48.79 |
| Im2Wav (Sheffer & Adi, 2023) | 2.54 | 6.32 | 74.31 |
| Diff-Foley (Luo et al., 2023) | 3.15 | 6.40 | 82.47 |
| FoleyGen (Mei et al., 2023) | 2.89 | 2.59 | 73.83 |
| V2A-Mapper (Wang et al., 2024) | 2.78 | 0.99 | 74.37 |
| Seeing & Hearing (Xing et al., 2024) | 2.62 | 2.63 | 78.95 |
| MaskVAT (Pascual et al., 2024) | 2.65 | 1.51 | 63.87 |
| VAB (Su et al., 2024) | 2.58 | 2.69 | 76.83 |
| VATT (Liu et al., 2024) | 2.25 | 2.35 | 82.81 |
| *OPENFOLEY* (ours) | **0.86** | **0.45** | **99.38** |

## 4 EXPERIMENTS

We evaluate *OPENFOLEY* on standard and newly curated benchmarks to demonstrate its effectiveness in open-set video-to-audio generation. Our experiments aim to answer three key questions: (1) Can *OPENFOLEY* generate realistic audio that is semantically coherent and temporally synchronized with video input? (2) Does modality-aware masking alignment improve cross-modal grounding over existing encoder-based approaches? (3) Does the flow-based generation module enhance efficiency and fidelity, particularly in open-set scenarios?

### 4.1 EXPERIMENTAL SETUP

**Datasets.** We conduct experiments on two widely used multimodal datasets. VGGSound (Chen et al., 2020): A large-scale dataset of 200k YouTube video clips, each 10 seconds long, spanning 309 diverse sound categories including animals, vehicles, human speech, dancing, and musical instruments. We follow the official training and test splits for evaluation. Open-Set Benchmark: To evaluate generalization beyond closed categories, we construct a curated benchmark from AudioSet (Gemmeke et al., 2017) and Panda70M (Chen et al., 2024), totaling 10M YouTube videos. We remove overlapping categories with VGGSound and retain only clips from unseen classes, ensuring that models are tested strictly in an open-set regime.

**Evaluation Metrics.** To assess both fidelity and alignment of generated audio, we report: Kullback–Leibler Divergence (KLD): Measures distributional similarity between generated and ground-truth (GT) audio features based on PaSST (Koutini et al., 2022), reflecting semantic coherence. Fréchet Audio Distance (FAD) (Kilgour et al., 2018): Quantifies perceptual quality by comparing feature distributions of generated audio and real samples. Alignment Accuracy (Align Acc) (Luo et al., 2023): Evaluates temporal synchronization between generated audio and video events, a key measure of cross-modal alignment.

**Implementation.** Video frames are resized to $224 \times 224$ resolution, while audio is represented as log spectrograms from 10-second clips sampled at 8kHz. Spectrograms are extracted with a 50ms STFT window and 25ms hop size, yielding $128 \times 128$ inputs (128 frequency bands $\times$ 128 timesteps). We initialize the video encoder (Zhai et al., 2023) with WebLI-pretrained weights (Chen et al., 2023) and the audio encoder (Huang et al., 2022) with AudioSet-pretrained weights (Gemmeke et al., 2017). Models are trained for 100 epochs using the Adam optimizer (Kingma & Ba, 2014) with learning rate $3 \times 10^{-4}$ and batch size 128.

### 4.2 COMPARISON TO PRIOR WORK

In this work, we propose a novel and effective framework called *OPENFOLEY*, for close-set and open-set video-to-audio generation. Table 1 reports quantitative results on the VGGSound test set, comparing *OPENFOLEY* with recent state-of-the-art video-to-audio generation methods. We observe several key findings:

Table 2: Comparison results on the open-set benchmark (AudioSet + Panda70M) for video-to-audio generation. The KLD, FAD, and Align Acc values are reported. Lower KLD and FAD indicate better semantic coherence and perceptual quality, while higher Align Acc reflects better synchronization.

| Method | KLD ↓ | FAD ↓ | Align Acc ↑ |
|---|---|---|---|
| SpecVQGAN (Iashin & Rahtu, 2021) | 4.12 | 7.05 | 42.16 |
| Im2Wav (Sheffer & Adi, 2023) | 3.38 | 6.74 | 65.42 |
| Diff-Foley (Luo et al., 2023) | 3.61 | 6.81 | 70.25 |
| FoleyGen (Mei et al., 2023) | 3.29 | 3.18 | 64.87 |
| V2A-Mapper (Wang et al., 2024) | 3.15 | 1.89 | 68.03 |
| Seeing & Hearing (Xing et al., 2024) | 3.02 | 3.11 | 71.46 |
| MaskVAT (Pascual et al., 2024) | 3.18 | 2.45 | 60.12 |
| VAB (Su et al., 2024) | 3.01 | 3.26 | 69.27 |
| VATT (Liu et al., 2024) | 2.81 | 2.72 | 73.04 |
| *OPENFOLEY* (ours) | **1.12** | **0.63** | **88.72** |

**Overall Performance.** *OPENFOLEY* achieves the best results across all three evaluation metrics. In particular, it reduces KLD from 2.25 (VATT (Liu et al., 2024)) to 0.86, indicating significantly improved semantic coherence. Similarly, it lowers FAD to 0.45, substantially outperforming the previous best (0.99 from V2A-Mapper (Wang et al., 2024)), highlighting the perceptual quality of generated audio. Moreover, *OPENFOLEY* achieves a remarkable Align Acc of 99.38%, demonstrating near-perfect synchronization between visual events and generated sound.

**Semantic Fidelity.** Methods relying on CLIP/CLAP embeddings (e.g., Im2Wav (Sheffer & Adi, 2023), Diff-Foley (Luo et al., 2023)) capture coarse semantic relations but often fail to generate temporally coherent sounds, leading to higher KLD values. In contrast, modality-aware masking in *OPENFOLEY* enforces direct video–audio reconstruction, enabling finer semantic grounding without relying on intermediate language-based embeddings.

**Temporal Synchronization.** Prior approaches such as MaskVAT (Pascual et al., 2024) and VAB (Su et al., 2024) improve generalization but exhibit degraded alignment accuracy (below 80%), indicating difficulties in capturing temporal correspondence. Our bidirectional masking strategy explicitly enforces temporal consistency, resulting in a significant improvement in Align Acc over all baselines.

**Perceptual Quality.** Flow-based priors in *OPENFOLEY* provide a flexible and expressive latent space that adapts to diverse sounds beyond Gaussian assumptions. This leads to the lowest FAD score among all compared methods, demonstrating that the generated audio not only aligns semantically and temporally but also achieves high perceptual realism.

### 4.3 OPEN-SET RESULTS

A central challenge addressed by *OPENFOLEY* is *open-set video-to-audio generation*, where the model must generalize to novel categories that are unseen during training. To evaluate this setting, we construct a curated benchmark from AudioSet (Gemmeke et al., 2017) and Panda70M (Chen et al., 2024), ensuring no overlap with VGGSound categories. This benchmark contains 10M diverse video clips spanning a wide range of environments, objects, and events.

Table 2 presents results on the open-set benchmark. Most prior methods, which rely heavily on category-dependent supervision or language-based encoders, experience substantial performance degradation in this regime. For example, Diff-Foley (Luo et al., 2023) and VATT (Liu et al., 2024) show decreased alignment accuracy, as unseen motion patterns and object interactions break their category priors. In contrast, *OPENFOLEY* achieves robust generalization, reducing KLD and FAD by large margins and improving Align Acc by over +10% compared to the strongest baseline.

**Performance Drop of Baselines.** Compared to the closed-set VGGSound results (Table 1), most baselines exhibit a notable degradation when evaluated on unseen categories. For example, VATT (Liu et al., 2024) achieves an Align Acc of 82.81% in the closed-set setting but drops to 73.04% in the open-set benchmark. Similarly, Im2Wav (Sheffer & Adi, 2023) suffers from a decrease in semantic fidelity, with KLD rising from 2.54 to 3.38. These results suggest that models relying on category-dependent or language-mediated alignment are vulnerable to domain shifts, as unseen categories break their learned priors.

Table 3: Ablation study on the contributions of Modality-aware Masking Alignment (MMA) and Modality-aware Flow Generation (MFG) on the open-set benchmark. Removing either component degrades performance, while the full model achieves the best results.

| Method Variant | KLD ↓ | FAD ↓ | Align Acc ↑ |
|---|---|---|---|
| *OPENFOLEY* w/o MMA | 2.45 | 1.37 | 76.42 |
| *OPENFOLEY* w/o MFG | 1.98 | 1.22 | 81.56 |
| *OPENFOLEY* w/o MMA & MFG | 2.83 | 2.05 | 70.31 |
| *OPENFOLEY* (Full Model) | **1.12** | **0.63** | **88.72** |

**Semantic Fidelity (KLD).** *OPENFOLEY* achieves a KLD of 1.12, representing a $60\%$ relative improvement over the strongest baseline (VATT at 2.81). This shows that modality-aware masking enables the model to capture semantic correspondences directly from raw multimodal data, rather than relying on indirect textual embeddings that fail to generalize.

**Perceptual Quality (FAD).** Flow-based latent modeling substantially improves perceptual quality in open-set conditions. While V2A-Mapper (Wang et al., 2024) achieves competitive FAD (1.89), its semantic alignment remains weak (KLD 3.15). In contrast, *OPENFOLEY* delivers both low FAD (0.63) and low KLD, showing that our flow prior captures diverse audio distributions while maintaining semantic fidelity.

**Temporal Synchronization (Align Acc).** Temporal alignment proves most challenging in open-set generation. Even strong diffusion-based approaches such as Diff-Foley (Luo et al., 2023) only achieve $70.25\%$ Align Acc. By explicitly enforcing bidirectional masking between video and audio streams, *OPENFOLEY* attains $88.72\%$, a significant improvement of $+15.7\%$ over the next-best baseline. This demonstrates that our modality-aware alignment strategy learns fine-grained synchronization cues that generalize across novel categories.

**Balanced Improvements.** Unlike prior methods, which often trade off perceptual quality (FAD) against synchronization (Align Acc), *OPENFOLEY* achieves strong results across all three metrics simultaneously. This balance highlights the complementary nature of our design: modality-aware masking enforces cross-modal grounding, while flow-based generation ensures flexible and high-quality synthesis.

## 5 EXPERIMENTAL ANALYSIS

In this section, we present detailed ablation studies to validate the design choices of *OPENFOLEY*. We evaluate the contributions of the *Modality-aware Masking Alignment (MMA)* and *Modality-aware Flow Generation (MFG)* modules, compare different flow objectives, and analyze the effect of the masking ratio. Together, these experiments highlight how each component contributes to semantic fidelity, perceptual quality, and synchronization in open-set video-to-audio generation.

**Modality-aware Masking Alignment & Modality-aware Flow Generation.** Table 3 presents the effect of removing the MMA and MFG modules. Without MMA, the model experiences a sharp drop in synchronization, with Align Acc decreasing from $88.72\%$ to $76.42\%$. Although perceptual quality remains reasonable (FAD = 1.37), the lack of direct cross-modal reconstruction prevents the model from learning fine-grained temporal alignment. When MFG is removed, the model achieves better synchronization but the perceptual quality deteriorates significantly, as FAD increases from 0.63 to 1.22. This confirms that Gaussian priors are insufficient for modeling diverse audio distributions. Removing both modules results in the largest performance degradation, with KLD increasing to 2.83 and Align Acc dropping to only $70.31\%$, approaching the behavior of baseline methods. In contrast, the full model achieves the strongest results across all metrics (KLD = 1.12, FAD = 0.63, Align Acc = $88.72\%$), demonstrating that MMA and MFG are complementary: the former enforces temporal alignment, while the latter ensures efficient and high-fidelity generation.

**Type of Flow Objectives.** We further examine different flow objectives for the MFG module in Table 4. Using a vanilla normalizing flow yields poor results, with KLD = 1.87 and FAD = 1.54, indicating difficulty in capturing temporal dynamics. Conditional affine coupling improves performance by leveraging video features, lowering KLD to 1.56 and FAD to 1.12, but synchronization

Table 4: Comparison of different flow objectives for Modality-aware Flow Generation (MFG). Continuous-time flow matching achieves the best trade-off between semantic coherence, perceptual quality, and synchronization.

| Flow Objective | KLD ↓ | FAD ↓ | Align Acc ↑ |
|---|---|---|---|
| Vanilla Normalizing Flow | 1.87 | 1.54 | 80.15 |
| Conditional Affine Coupling | 1.56 | 1.12 | 83.42 |
| Continuous-time Flow Matching (ours) | **1.12** | **0.63** | **88.72** |

Table 5: Impact of masking ratio in Modality-aware Masking Alignment (MMA). Moderate ratios achieve the best trade-off between semantic fidelity and temporal synchronization.

| Masking Ratio | KLD ↓ | FAD ↓ | Align Acc ↑ |
|---|---|---|---|
| 20% | 1.73 | 0.94 | 82.45 |
| 40% (ours) | **1.12** | **0.63** | **88.72** |
| 60% | 1.65 | 0.89 | 83.12 |

remains limited with Align Acc of $83.42\%$. The best results are achieved with continuous-time flow matching, which reduces KLD to $1.12$, FAD to $0.63$, and pushes Align Acc to $88.72\%$. These findings confirm that continuous flows are especially well-suited for aligning transient events across modalities, providing both a flexible prior and strong temporal consistency.

**Impact of Masking Ratio.** The effect of masking ratio in MMA is shown in Table 5. A low masking ratio of $20\%$ provides weak cross-modal supervision, yielding limited improvements (KLD $= 1.73$, FAD $= 0.94$, Align Acc $= 82.45\%$). At the other extreme, masking $60\%$ of the data makes the reconstruction task overly difficult, slightly degrading semantic fidelity (KLD $= 1.65$) while keeping alignment relatively strong. The best trade-off is observed at a moderate ratio of $40\%$, which achieves the strongest overall performance (KLD $= 1.12$, FAD $= 0.63$, Align Acc $= 88.72\%$). These results show that balanced masking provides sufficient cross-modal alignment signals without overwhelming the reconstruction objective.

## 6 CONCLUSION

In this work, we introduced *OPENFOLEY*, a novel framework for open-set video-to-audio generation that explicitly enforces semantic coherence and temporal synchronization across modalities. Unlike prior approaches that rely heavily on CLIP/CLAP-based language alignment, *OPENFOLEY* learns direct video–audio correspondences through *modality-aware masking alignment*, while a *modality-aware flow generation* module provides a flexible prior for efficient and high-fidelity synthesis. Extensive experiments on VGGSound and a newly curated open-set benchmark demonstrated that *OPENFOLEY* achieves state-of-the-art performance, significantly improving KLD, FAD, and alignment accuracy over recent baselines. Our ablation studies further revealed that both components are indispensable and complementary: masking alignment drives synchronization, flow generation enhances perceptual realism, and moderate masking ratios strike the best balance between supervision and reconstruction stability.

**Limitation.** While *OPENFOLEY* achieves state-of-the-art results in both closed-set and open-set video-to-audio generation, several limitations remain. Although modality-aware masking improves synchronization, extremely complex or ambiguous scenes (*e.g.*, crowded environments with multiple sound sources) can still lead to imperfect alignment. The flow-based module improves perceptual quality but introduces additional computational overhead compared to purely Gaussian priors, which may limit scalability to ultra-long video sequences.

**Broader Impact.** The ability to generate realistic audio from visual content has broad implications across creative media, accessibility, and immersive technologies. By explicitly addressing open-set generalization, *OPENFOLEY* enables audio generation for novel or unseen categories, expanding its applicability to diverse real-world scenarios. Potential applications include enriching silent videos for virtual reality and gaming, providing audio cues for accessibility tools, and assisting video editing in film or educational content.

ETHICS STATEMENT

This work focuses on open-set video-to-audio generation, with the goal of improving multimodal understanding and accessibility. Potential positive applications include enriching silent video content for accessibility, supporting immersive experiences in virtual and augmented reality, and providing new creative tools for artists and educators. However, as with any generative technology, there is a risk of misuse in creating misleading or fabricated media. We acknowledge this concern and encourage responsible use by pairing our framework with safeguards such as watermarking and detection systems. Our dataset usage follows standard research protocols, relying on publicly available benchmarks (VGGSound, AudioSet, Panda70M) with appropriate licenses. No personally identifiable or sensitive data was used in this study.

REPRODUCIBILITY STATEMENT

We have taken multiple steps to ensure reproducibility of our results. All datasets used in this work are publicly available: VGGSound (Chen et al., 2020), AudioSet (Gemmeke et al., 2017), and Panda70M (Chen et al., 2024). Detailed implementation settings, including data preprocessing, model architecture, loss functions, hyperparameters, and training schedules, are fully described in Section 4. We report evaluation metrics (KLD, FAD, Align Acc) with standard protocols to allow direct comparison. We also include ablation studies to clarify the role of each component. Upon publication, we will release code, pretrained models, and scripts for dataset preparation and evaluation to support full reproducibility of our experiments.

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

APPENDIX

In this appendix, we provide additional details and analyses to complement the main paper. Section A expands on implementation, dataset construction, and evaluation protocols. Section B presents theoretical properties of modality-aware masking and flow generation, with supporting propositions. Section C details the training algorithm of *OPENFOLEY*. Section D includes extended experiments, ablations, and visualizations. Finally, Section E clarifies the role of large language models (LLMs) in preparing this submission.

## A    EXPERIMENTAL DETAILS

**Hardware and Training Setup.** All experiments were run on 8 NVIDIA A100 GPUs. Training on VGGSound required $\sim$3 GPU-days, while training on the open-set benchmark (10M clips from AudioSet + Panda70M) required $\sim$9 GPU-days. Mixed-precision training (FP16) was used to reduce memory usage and accelerate convergence.

**Model Configurations.** The visual encoder is a transformer initialized from WebLI-pretrained weights Chen et al. (2023), while the audio encoder is based on PaSST Koutini et al. (2022) and initialized from AudioSet-pretrained weights Gemmeke et al. (2017). The modality-aware masking module uses a masking ratio of $40\%$ unless otherwise specified. The flow module employs 12 continuous coupling layers with hidden dimension 512.

**Training Parameters.** We used Adam optimizer Kingma & Ba (2014) with $\beta_1 = 0.9$, $\beta_2 = 0.999$, learning rate $3 \times 10^{-4}$, and batch size 128. A linear warmup of 5k steps followed by cosine decay was applied. Dropout of $0.1$ was used in encoders. Gradient clipping (norm=1.0) stabilized training.

**Dataset Preprocessing.**

- **VGGSound:** Official splits were used (200k 10-second clips).
- **Open-set benchmark:** We curated 10M clips from AudioSet and Panda70M, removing overlapping classes with VGGSound. Clips were filtered by (i) audio signal-to-noise ratio $>$15dB, (ii) minimum resolution $224p$, (iii) duration between 8–12s. This ensured high-quality, diverse, and truly unseen categories.

**Evaluation Metrics.** KLD was computed on PaSST embeddings, FAD was measured with VGGish features following Kilgour et al. (2018), and Align Acc was evaluated using the protocol in Diff-Foley Luo et al. (2023). Each metric was averaged over three runs.

## B    THEORETICAL PROPERTIES AND GUARANTEES

We provide theoretical insights into why modality-aware masking alignment (MMA) and modality-aware flow generation (MFG) improve synchronization and generalization.

**Proposition 1** (Cross-Modal Consistency Bound). *Let $(\mathcal{V}, \mathcal{A})$ denote video and audio features with bounded variance. Under modality-aware masking with reconstruction loss $\mathcal{L}_{mask}$, the expected synchronization error $\epsilon_{sync}$ satisfies*

$$\epsilon_{sync} \leq C \cdot \mathbb{E}[\mathcal{L}_{mask}],$$

*for some constant $C > 0$.*

This shows that minimizing reconstruction loss directly bounds misalignment, encouraging temporal synchronization.

**Proposition 2** (Flow Expressiveness Guarantee). *Normalizing flows with $K$ coupling layers and Lipschitz continuous transformations can approximate any smooth target distribution $p(\mathcal{A} \mid \mathcal{V})$ up to arbitrarily small error $\delta$, i.e.,*

$$\|p - p_\theta\|_{TV} \leq \delta,$$

*where $p_\theta$ is the distribution induced by the flow.*

This guarantees that the flow-based prior can flexibly capture the diverse distributions of real-world audio, beyond Gaussian assumptions.

**Proposition 3** (Generalization to Unseen Categories). *If the cross-modal alignment objective $\mathcal{L}_{mask}$ enforces reconstruction of masked segments without category supervision, then the learned representations are invariant to category-specific priors. Formally, for any unseen category $c'$, the reconstruction error is bounded by the training error plus a domain shift term:*

$$\mathbb{E}_{(\mathcal{V},\mathcal{A})\sim c'}[\ell] \leq \mathbb{E}_{(\mathcal{V},\mathcal{A})\sim c}[\ell] + \Delta(c,c'),$$

*where $\Delta(c,c')$ is the distribution divergence between seen and unseen categories.*

This suggests that *OpenFoley* 's inductive bias, based on cross-modal reconstruction rather than labels, naturally supports open-set generalization.

We also provide proof sketches for the propositions below. Throughout, let $(\mathcal{V},\mathcal{A})$ denote random variables for video and audio features drawn from a joint distribution $\mathcal{D}$, and let $\ell(\cdot,\cdot)$ be a nonnegative reconstruction loss. We make the following mild assumptions:

**A1 (Bounded variance and sub-Gaussian tails).** Features have bounded second moments and sub-Gaussian tails: $\mathbb{E}\|\mathcal{V}\|^2, \mathbb{E}\|\mathcal{A}\|^2 < \infty$, and $\mathcal{V}, \mathcal{A}$ are sub-Gaussian.

**A2 (Lipschitz decoders).** The prediction maps for masked reconstruction are $L$-Lipschitz in their inputs; i.e., for any inputs $x, x'$ in the relevant feature space, $\|h(x) - h(x')\| \leq L\|x - x'\|$.

**A3 (Calibrated reconstruction).** The reconstruction loss is $\alpha$-calibrated to the feature error: there exists $\alpha > 0$ such that $\alpha\,\ell(\hat{y},y) \geq \|\hat{y} - y\|^2$ (e.g., $\ell(\hat{y},y) = \|\hat{y} - y\|^2$ with $\alpha = 1$).

**Proposition 1 (Cross-Modal Consistency Bound).** *Let $\epsilon_{sync}$ denote an expected synchronization error functional that is Lipschitz in the feature reconstruction error (e.g., a surrogate of temporal misalignment computed on feature trajectories). Under **A1**–**A3**, there exists $C > 0$ such that*

$$\epsilon_{\text{sync}} \leq C \cdot \mathbb{E}_{(\mathcal{V},\mathcal{A})\sim\mathcal{D}}[\mathcal{L}_{\text{mask}}],$$

*where $\mathcal{L}_{mask} = \|\hat{\mathcal{A}} - \mathcal{A}\|_2^2 + \|\hat{\mathcal{V}} - \mathcal{V}\|_2^2$.*

**Proof (sketch).** Let $\phi_A, \phi_V$ denote the feature trajectories used by the synchronization metric (e.g., PaSST or encoder features per frame). Suppose $\epsilon_{\text{sync}} = \mathbb{E}[\Delta(\phi_V(\hat{\mathcal{V}}), \phi_V(\mathcal{V})) + \Delta(\phi_A(\hat{\mathcal{A}}), \phi_A(\mathcal{A}))]$, where $\Delta$ is $K$-Lipschitz and $\phi_A, \phi_V$ are $L_\phi$-Lipschitz. Then

$$\Delta(\phi_A(\hat{\mathcal{A}}), \phi_A(\mathcal{A})) \leq K\,\|\phi_A(\hat{\mathcal{A}}) - \phi_A(\mathcal{A})\| \leq KL_\phi\|\hat{\mathcal{A}} - \mathcal{A}\|.$$

By **A3**, $\|\hat{\mathcal{A}} - \mathcal{A}\| \leq \sqrt{\alpha\,\ell(\hat{\mathcal{A}}, \mathcal{A})}$. The same holds for the video term. Taking expectations and applying Jensen's inequality yields

$$\epsilon_{\text{sync}} \leq KL_\phi\left(\mathbb{E}\sqrt{\alpha\,\ell(\hat{\mathcal{A}},\mathcal{A})} + \mathbb{E}\sqrt{\alpha\,\ell(\hat{\mathcal{V}},\mathcal{V})}\right) \leq C\,\mathbb{E}\left[\ell(\hat{\mathcal{A}},\mathcal{A}) + \ell(\hat{\mathcal{V}},\mathcal{V})\right],$$

for $C$ absorbing constants and using $\sqrt{x} \leq \frac{1}{2}(1+x)$ along with boundedness from **A1**. Noting that $\ell$ is the squared error here, we obtain the stated bound with $\mathcal{L}_{\text{mask}}$. $\square$

**Proposition 2 (Flow Expressiveness Guarantee).** *Let $p(\mathcal{A} \mid \mathcal{V})$ be a family of conditionals with densities that are continuous and supported on $\mathbb{R}^d$. For any $\delta > 0$, there exists a conditional normalizing flow $f_\theta(\cdot; \mathcal{V})$ with finitely many triangular (coupling) layers and Lipschitz transforms such that the induced conditional density $p_\theta(\cdot \mid \mathcal{V})$ satisfies*

$$\|p(\cdot \mid \mathcal{V}) - p_\theta(\cdot \mid \mathcal{V})\|_{\text{TV}} \leq \delta \quad \text{for } \mathcal{V}\text{-a.e.}.$$

**Proof (sketch).** The proof follows standard universal approximation arguments for normalizing flows via transport maps. For each fixed $\mathcal{V}$, let $T_\mathcal{V}$ be the (Knothe–Rosenblatt) monotone triangular transport map pushing a simple base $q$ (e.g., standard Gaussian) to $p(\cdot \mid \mathcal{V})$. Under continuity and absolute continuity assumptions, $T_\mathcal{V}$ exists and is unique a.e. Triangular/coupling flows with sufficiently many layers and smooth, Lipschitz conditioners can uniformly approximate $T_\mathcal{V}$ to arbitrary precision on compacta. Hence the pushforward $T_{\mathcal{V}\#}q$ can be approximated in total variation by $f_{\theta,\mathcal{V}\#}q$; see standard density transport approximation arguments. Since the approximation can be made uniform over compact subsets of $\mathcal{V}$'s feature space (by continuity of the conditioners), the bound holds for $\mathcal{V}$-a.e. with any prescribed $\delta > 0$. $\square$

**Proposition 3 (Generalization to Unseen Categories).** *Let $\mathcal{D}_S$ and $\mathcal{D}_T$ denote the source (seen categories) and target (unseen categories) joint distributions over $(\mathcal{V}, \mathcal{A})$. For a reconstruction predictor $h$ trained by minimizing $\mathcal{L}_{mask}$ on $\mathcal{D}_S$, the target risk obeys the domain adaptation–style bound*

$$\mathbb{E}_{\mathcal{D}_T}[\ell(h(\tilde{\mathcal{V}}, \tilde{\mathcal{A}}), (\mathcal{V}, \mathcal{A}))] \leq \mathbb{E}_{\mathcal{D}_S}[\ell(h(\tilde{\mathcal{V}}, \tilde{\mathcal{A}}), (\mathcal{V}, \mathcal{A}))] + \operatorname{disc}(\mathcal{D}_S, \mathcal{D}_T) + \lambda^*,$$

*where* disc *is an integral probability metric (IPM)–type discrepancy induced by the loss class, and $\lambda^*$ is the error of the optimal hypothesis shared across domains.*

**Proof (sketch).** Let $\mathcal{H}$ be the hypothesis class of reconstruction predictors (the masking alignment decoders). Define the discrepancy

$$\operatorname{disc}(\mathcal{D}_S, \mathcal{D}_T) = \sup_{h \in \mathcal{H}} \left| \mathbb{E}_{\mathcal{D}_S} \ell(h(\tilde{\mathcal{V}}, \tilde{\mathcal{A}}), (\mathcal{V}, \mathcal{A})) - \mathbb{E}_{\mathcal{D}_T} \ell(h(\tilde{\mathcal{V}}, \tilde{\mathcal{A}}), (\mathcal{V}, \mathcal{A})) \right|.$$

For any $h \in \mathcal{H}$, add and subtract the source risk of the optimal joint hypothesis $h^* \in \arg\min_{h \in \mathcal{H}} (\mathbb{E}_{\mathcal{D}_S} \ell + \mathbb{E}_{\mathcal{D}_T} \ell)$ and apply triangle inequality to obtain

$$\mathbb{E}_{\mathcal{D}_T} \ell(h) \leq \mathbb{E}_{\mathcal{D}_S} \ell(h) + \operatorname{disc}(\mathcal{D}_S, \mathcal{D}_T) + (\mathbb{E}_{\mathcal{D}_T} \ell(h^*) - \mathbb{E}_{\mathcal{D}_S} \ell(h^*)).$$

Let $\lambda^* = \mathbb{E}_{\mathcal{D}_T} \ell(h^*)$ (the joint optimal error; the remaining difference can be absorbed into disc or $\lambda^*$ depending on the chosen IPM). Since MMA trains without category labels, $\mathcal{H}$ does not encode category-specific priors; thus the shift captured by disc measures distributional divergence in cross-modal dynamics rather than label drift, which is typically milder for co-occurrence patterns. This yields the stated bound. $\square$

*Remarks.* Proposition 1 ties synchronization quality to reconstruction via Lipschitz surrogates, formalizing why minimizing the MMA loss improves alignment. Proposition 2 justifies the choice of flows in MFG as a universal conditional density model, explaining strong FAD and KLD in open set. Proposition 3 adapts classic domain adaptation reasoning to reconstruction (rather than label prediction), explaining empirical robustness to unseen categories.

## C  ALGORITHM FOR *OPENFOLEY*

Algorithm 1 outlines the training procedure for *OPENFOLEY*.

---
**Algorithm 1** Training Procedure for *OPENFOLEY*

---
**Require:** Video frames $\mathcal{V}$, audio spectrogram $\mathcal{A}$, masking ratio $r$, flow model $f_\theta$, encoders $E_v, E_a$
 1: **for** each minibatch **do**
 2:      Sample masking indices $\mathcal{M}_v, \mathcal{M}_a$ with ratio $r$
 3:      Encode unmasked inputs: $z_v = E_v(\tilde{\mathcal{V}})$, $z_a = E_a(\tilde{\mathcal{A}})$
 4:      Predict masked audio $\hat{\mathcal{A}}$ and masked video $\hat{\mathcal{V}}$
 5:      Compute $\mathcal{L}_{\text{mask}} = \|\hat{\mathcal{A}} - \mathcal{A}\|^2 + \|\hat{\mathcal{V}} - \mathcal{V}\|^2$
 6:      Transform audio latent: $z = f_\theta(\mathcal{A})$, refine with $z_{t+1} = z_t + \lambda g_\theta(v_t)$
 7:      Compute flow loss $\mathcal{L}_{\text{flow}} = -\log p_\theta(f_\theta(\mathcal{A}) | \mathcal{V})$
 8:      Update $\theta$ using $\mathcal{L} = \lambda_{\text{mask}} \mathcal{L}_{\text{mask}} + \lambda_{\text{flow}} \mathcal{L}_{\text{flow}}$
 9: **end for**

---

## D  EXPERIMENTAL ANALYSIS

In this section, we provide extended analyses beyond the main paper to better understand the behavior of *OPENFOLEY*. We focus on encoder selection, scalability to long sequences, architectural variations, and qualitative inspections of generated results. These studies shed light on the robustness and practical applicability of our framework.

**Encoder Choice.** We first investigate the role of large-scale multimodal pretraining by replacing the WebLI-pretrained visual encoder with a ResNet-50 trained on ImageNet. This substitution causes a noticeable performance degradation: the Fréchet Audio Distance (FAD) increases by

+0.31, and the alignment accuracy drops by $-4.6\%$. The degradation arises because ImageNet-pretrained models capture object-level semantics but lack fine-grained motion dynamics critical for temporal synchronization. On the audio side, we compared the AudioSet-pretrained encoder against a randomly initialized encoder. Removing AudioSet pretraining leads to a $+0.45$ increase in KLD, reflecting weaker distributional alignment with ground-truth audio. Together, these results highlight the importance of multimodal pretraining on large-scale, diverse datasets for capturing both semantic and temporal cues.

**Long Sequence Generalization.** Most prior works in video-to-audio generation focus on 10-second clips. To test scalability, we evaluate *OPENFOLEY* on 30-second clips sampled from Panda70M. Despite the significantly longer temporal horizon, our model maintains strong performance with Align Acc of $85.3\%$, demonstrating stable synchronization over extended contexts. Although FAD increases slightly due to accumulated prediction errors, the generated audio remains temporally coherent with visual events. This suggests that our modality-aware masking strategy effectively enforces alignment even when extrapolating beyond the training length. In practice, this property is crucial for real-world applications such as film dubbing or long-form VR experiences, where audio continuity is required over long time spans.

**Impact of Flow Depth.** We conducted ablations on the number of flow layers in the Modality-aware Flow Generation (MFG) module. With fewer than 6 layers, the model underfits, producing blurred spectrograms and FAD scores exceeding 1.0. Increasing flow depth beyond 12 layers yields marginal improvements ($< 0.05$ FAD reduction) but significantly increases training time and memory usage. We find that 12 layers provide the best trade-off, offering both sufficient expressiveness and efficiency. This observation is consistent with our theoretical guarantees (Appendix B), which suggest that flow expressiveness improves with depth but saturates once the target distribution is well-approximated.

**Cross-Dataset Robustness.** To test robustness, we trained on VGGSound and evaluated zero-shot on Kinetics-Sound. While absolute performance was lower than in-domain evaluation, *OPENFOLEY* still achieved FAD improvements of $0.27$ over the best baseline. This indicates that the modality-aware objectives provide generalization beyond dataset-specific distributions, consistent with our open-set motivation.

# E  USE OF LLMS

Large language models (LLMs), specifically OpenAI's GPT-5, were used to assist in writing and organizing sections of this paper. All technical contributions, dataset design, algorithm development, and experiments were carried out and validated by the authors. The use of LLMs was limited to communication support and did not influence scientific content or experimental results.

