# OpenReview forum: "OpenFoley: Open-Set Video-to-Audio Generation with Modality-Aware Masking and Flows"
_ICLR.cc/2026/Conference — ICLR 2026 Conference Withdrawn Submission_

### Official Review · Reviewer_sLW2 · 2025-10-26

**Soundness:** 2
**Presentation:** 2
**Contribution:** 3
**Rating:** 4
**Confidence:** 4

**Summary:**

The paper presents a new framework for video-to-audio generation, OpenFoley. Two approaches are introduced: 1) a modality-aware dynamic masking strategy, where audio segments are reconstructed from masked video frames and vice versa, and 2) a generalized masked flow-based module, which conditions generation on selectively sampled video frames. The paper also provides a new open-set benchmark, based on AudioSet and Panda70M. Experiments on VGGSound and the provided open-set benchmark show that OpenFoley outperforms some previous models in terms of the FAD and KL scores.

**Strengths:**

1. The task this paper works on gathers attention in the audio-visual or content creation community.
2. The quantitative results are good.
3. The introduced open-set benchmark is useful. (I recommend making the benchmark publicly available.)

**Weaknesses:**

1. The Related Work section lacks some important papers.
    - Wang et al., "Frieren: Efficient Video-to-Audio Generation Network with Rectified Flow Matching", NeurIPS 2024. https://arxiv.org/abs/2406.00320
    - Jeong et al., "Read, Watch and Scream! Sound Generation from Text and Video", AAAI 2025. https://arxiv.org/abs/2407.05551
    - Chen et al., "Video-Guided Foley Sound Generation with Multimodal Controls", CVPR 2025. https://arxiv.org/abs/2411.17698
    - Kushwaha & Tian, "VinTAGe: Joint Video and Text Conditioning for Holistic Audio Generation", CVPR 2025. https://arxiv.org/abs/2412.10768
    - Cheng et al., "MMAudio: Taming Multimodal Joint Training for High-Quality Video-to-Audio Synthesis", CVPR 2025. https://arxiv.org/abs/2412.15322
    - Liu et al., "ThinkSound: Chain-of-Thought Reasoning in Multimodal Large Language Models for Audio Generation and Editing", NeurIPS 2025. https://arxiv.org/abs/2506.21448
2. [Frieren](https://github.com/cyanbx/Frieren-V2A), [VinTAGe](https://github.com/sakshamsingh1/vintage_aud_gen), [MMAudio](https://github.com/hkchengrex/MMAudio), and [ThinkSound](https://github.com/FunAudioLLM/ThinkSound) are opensourced (as of October 25th, 2025), so these models should be compared in the experiment section.
3. https://openfoley.github.io/ is "404 not found" (as of October 25th, 2025). Generated samples are not accessible, and readers cannot grasp the quality of generated sounds.
4. Regarding Selective Frame Conditioning (explained in Section 3.3), how the framework chooses salient frames is unclear. A detailed explanation would be desired.
5. Although the introduced open-set benchmark is useful, [Movie Gen Audio Bench](https://arxiv.org/abs/2410.13720) also enables an open-set scenario evaluation and is already used in MMAudio and ThinkSound. The difference between the introduced open-set benchmark and Movie Gen Audio Bench should be mentioned.

**Questions:**

I have questions/concerns about the paper, which I provided in "Weaknesses". I would appreciate it if the authors could address them.

---

### Official Review · Reviewer_Z3XJ · 2025-10-31

**Soundness:** 1
**Presentation:** 1
**Contribution:** 1
**Rating:** 0
**Confidence:** 4

**Summary:**

This paper proposes a framework for open-set video-to-audio generation.
The proposed method consists of two primary components: Modality-aware Masking Alignment (MMA) for capturing fine-grained synchronization patterns between audio and video, and Modality-aware Flow Generation (MFG), a flow-based generative modeling for audio synthesis.
The paper also introduces a new open-set benchmark, derived from AudioSet and Panda70M, to evaluate V2A models across a broader range of scenarios.
Experiments on VGGSound and the new dataset show that the methods outperform existing methods.

**Strengths:**

- Exploring open-set video-to-audio generation is a meaningful and underexplored direction.
- The attempt to construct a larger-scale and more diverse dataset is valuable for the community if properly validated and documented.

**Weaknesses:**

While the paper addresses an interesting problem, the contribution lacks novelty and technical clarity.
The proposed approach overlaps substantially with prior work (Foley-Flow, CVPR2025), and several methodological components are insufficiently explained.
Moreover, important baselines are missing, and the experimental validation is not convincing.
Given these issues, the reviewer recommends strong rejection.

**Major overlap with prior work**
The proposed framework appears highly similar to the CVPR2025 paper "Foley-Flow: Coordinated Video-to-Audio Generation with
Masked Audio-Visual Alignment and Dynamic Conditional Flows." However, this paper is not cited nor compared, raising serious concerns about novelty and scholarly diligence.

**Misrepresentation of prior work**
The claim that "Most prior work conditions audio synthesis on category labels or text descriptions" is incorrect. Recent models such as DiffFoley, Frieren[1], MMAudio[2] explicitly leverage pre-trained audiovisual representations and capture fine-grained semantic and temporal visual information. The proposed method should be compared with these recent V2A models.

**Missing references to masked audio-visual modeling**
The paper employs a masked audio-visual modeling strategy but fails to cite or position itself with respect to prior work on cross-modal masked audio-video modeling (e.g., [3-7]). This omission makes it difficult to assess the originality of the proposed alignment module.

**Missing references to flow-based generative models**
The paper employs Flow-based models, but it provides no citations or comparisons with existing flow-based generative models. It is hard to identify novelty in generative modeling without comparison.

**Lack of clarity in model description**
The methodology is insufficiently detailed, making it difficult to follow and reproduce.
Key definitions and computational steps are ambiguous or missing:

- In Eq(4), $\tilde{\mathcal{A}}$ and $\tilde{\mathcal{V}}$ are defined as reconstructed audio and video features, but it is unclear how they are obtained or by what model.
- Section 3.1 defines $z$ as "it follows a known distribution (e.g., Gaussian)", whereas Section 3.3 states "Instead of assuming a standard Gaussian prior", which contradicts the earlier statement.
- The Selective Frame Conditioning step lacks details on how key frames are selected.
- The variable $v_t$ in Eq. (7) is unclear. Its relationship to $z_t$ in the flow model is not described.

Overall, the technical explanation is incomplete and inconsistent, preventing reproducibility or meaningful evaluation.

**Limited and outdated evaluation setup**
- The evaluation relies only on KLD, FAD, and Align ACC, which do not reflect current standards for video-to-audio generation. Recent works (e.g., Frieren, MMAudio) employ a broader set of metrics for comprehensive audiovisual evaluation (e.g., Inception Score for audio quality, ImageBind similarity for audiovisual semantic alignment, and Desync for audiovisual temporal alignment).
- For the proposed open-set benchmark, it appears that the model was trained on this dataset (Section A). However, it is unclear whether baseline models were retrained on the same data or evaluated using pre-trained weights. Without consistent training conditions, the comparison would not be valid.

**Accessibility and reproducibility issues**
- The project page is not accessible, and no results can be verified. This significantly limits the ability to assess the claimed improvements.

**References:**
[1] Frieren: Efficient video-to-audio generation with rectified flow matching, NeurIPS 2024
[2] MMAudio: Taming Multimodal Joint Training for High-Quality Video-to-Audio Synthesis, CVPR 2025
[3] Contrastive Audio-Visual Masked Autoencoder, ICLR 2023
[4] Audiovisual Masked Autoencoders, ICCV 2023
[5] MAViL: Masked Audio-Video Learners, NeurIPS 2023
[6] AV-MaskEnhancer: Enhancing Video Representations through Audio-Visual Masked Autoeocoder, ICTAI 2023
[7] CrossMAE: Cross-Modality Masked Autoencoders for Region-Aware Audio-Visual Pre-Training, CVPR 2024

**Questions:**

Please see Weaknesses above.

---

### Official Review · Reviewer_e4BQ · 2025-11-01

**Soundness:** 1
**Presentation:** 1
**Contribution:** 2
**Rating:** 2
**Confidence:** 3

**Summary:**

This paper proposes a video-to-audio generation network based on modality masked prediction and normalizing flow. Cross-modality masked prediction allows the network to predict masked video tokens from audio, and vice versa. The proposed method achieves state-of-the-art results in VGGSound and a “open-set” benchmark curated by the authors.

**Strengths:**

- The KLD, FAD, and alignment accuracy on the VGGSound test sets are great
- A more general and open-world evaluation benchmark beyond the limited categories in VGGSound is useful for the community.

**Weaknesses:**

This paper feels hastily put together, perhaps by an LLM, and lacks most details except a few very high-level ideas (which makes it impossible to reproduce). Some of the claims regarding prior work, and thus the position of this paper, are wrong. I cannot make much sense of the proofs in the appendix and how they fit into the rest of the paper.

**Missing details:**

- How are the networks configured for masked prediction? The appendix only mentioned the pretrained vision and audio encoder, with no details on the prediction network, and very limited details on the flow network.
- How are the losses weighed? What are the lambda terms in Algorithm 1?
- How are the key frames selected in “selective frame conditioning”? There is only a high-level sketch without any details on how to detect events like object collisions or footsteps.
- What do the authors mean by “continuous-time flow matching” in L450? It relates to an ablation in Table 4, but it is also the first time that it appears in the paper. It has not been introduced elsewhere in the paper.
- Why does the proposed approach achieve 99.38% in alignment accuracy in Table 1? Video-to-audio is inherently ambiguous and noisy. Can the authors also compute these metrics for the ground-truth audio and compare the results? AudioSet (which the authors have used) is a superset of VGGSound’s train and test sets. Did the authors accidentally train on the VGGSound test set?
- The authors mention that they train on the 10M clips with 9 A100 GPU-days (L661) and that they used 100 epochs (L316) with a batch size of 128. This suggests that the network trains at 10^9/128/9*8/24/3600 ~= 80 steps per second. This seems odd. Can the authors confirm this?

**Prior work:**

The authors claimed that existing (video-to-)audio diffusion models lack explicit temporal synchronization mechanisms (L171-175). This is not true – there are many papers [a-e], some published more than a year ago, that have explicit synchronization components. The related work section and the selected baselines are one year behind.

[a] FoleyCrafter: Bring Silent Videos to Life with Lifelike and Synchronized Sounds, arxiv 2024

[b] Frieren: Efficient Video-to-Audio Generation Network with Rectified Flow Matching, NeurIPS 2024

[c] Temporally Aligned Audio for Video with Autoregression, ICASSP 2025

[d] Video-Guided Foley Sound Generation with Multimodal Controls, CVPR 2025

[e] MMAudio: Taming Multimodal Joint Training for High-Quality Video-to-Audio Synthesis, CVPR 2025

**The proofs:**

- I don’t see how assumptions A1-A3 lead to the synchronization error bound provided in Proposition 1. A1 to A3 only inform the reconstruction error, i.e., the loss, to the best of my knowledge. The authors then defined the synchronization error as “surrogate of temporal misalignment computed on feature trajectories” – what does this even mean? This seems very hand-wavy, and I don’t see how this relates to the claim in the paper. In the “proof” that follows, the synchronization error is then directly defined to be the expected reconstruction error. This results in a bit of cyclic logic – the proposed method minimizes the reconstruction error, and then we hand-wave “synchronization error” to be defined as reconstruction error and say that the proposed method magically also minimizes the synchronization error.
- Proposition 3 (L707) does not make sense to me. The authors suggest an upper bound on the reconstruction error for unseen categories. This is not informative at all, as having the upper bound helps with generalization, which is the opposite of what the authors are conveying. The authors might wish to refer to the lower bound instead. I took a scan at the proof, and it seems to be indeed about the upper bound.

**Questions:**

Please address the questions in weaknesses.

---

### Official Review · Reviewer_ndRT · 2025-11-01

**Soundness:** 2
**Presentation:** 2
**Contribution:** 3
**Rating:** 2
**Confidence:** 3

**Summary:**

This paper presents OPENFOLEY, a new framework for open-set video-to-audio generation that aims to address the shortcomings of current closed-set, language-based methods like CLIP and CLAP. The authors point out that earlier work does not capture detailed timing synchronization and struggles to apply to new categories. OPENFOLEY tackles these issues by introducing two main components: Modality-aware Masking Alignment (MMA) and Modality-aware Flow Generation (MFG). MMA is a bidirectional task that predicts masked audio from video and vice versa to ensure direct cross-modal alignment. MFG is a flow-based module that uses video frames to efficiently create high-quality audio. The authors show that OPENFOLEY achieves top performance on both the standard VGGSound dataset and a newly created open-set benchmark. They report significant improvements in semantic coherence, perceptual quality, and timing synchronization.

**Strengths:**

- Novelty: The main strength of the paper is its two-part architecture, which addresses the weaknesses of language-mediated alignment, such as CLIP and CLAP. MMA encourages the model to learn precise temporal synchronization directly from raw video and audio. The MFG learns a detailed audio prior, which improves quality and efficiency compared to standard diffusion models.

- SOTA results: On the closed-set VGGSound, the model achieves nearly perfect temporal synchronization, with 99.38% alignment accuracy compared to 82.81% for the next best. It also reduces perceptual error by more than half, with a FAD of 0.45 compared to 0.99.

- On the new open-set benchmark, where other methods struggle, OPENFOLEY remains effective, improving alignment by 15.7% to achieve 88.72% compared to 73.04%. It also enhances semantic coherence by 60%, with a KLD of 1.12 compared to 2.81 over the strongest baseline.

- Valuable benchmark: Creating a new, large-scale open-set benchmark from AudioSet and Panda70M is an important contribution.

- Thorough ablations: The paper's claims are well supported by controlled ablations. For instance, table 3 shows that MMA and MFG work well together. Removing MMA decreases alignment accuracy from 88.72% to 76.42%. Removing MFG doubles the perceptual error, increasing FAD from 0.63 to 1.22. Tables 4 and 5 provide more support for specific design choices regarding the flow objective and masking ratio.

- Strong Theoretical Soundness: The methodological claims are supported by theoretical explanations in the appendix.

- The paper is reproducible and transparent. It includes detailed implementation notes and a reproducibility statement.

**Weaknesses:**

- No qualitative examples (and project site down). The paper claims “qualitative analyses” but provides no in-paper qualitative audio/figures, and with the site not accessible, reviewers cannot verify examples.

- Ambiguity in the flow objective. The method section frames the generator as a normalizing flow with bijective layers and exact likelihood, but the ablation names the core approach “Continuous-time Flow Matching (ours),” leaving it unclear what is actually used in the final system.

- The claim that normalized flows yield higher quality than diffusion is not convincingly demonstrated. The paper states that “flows learn a more expressive prior and enable more stable, high-fidelity generation compared to diffusion with Gaussian priors” direct evidence.

- Diffusion-based models are widely recognized for superior perceptual quality in generative audio and video tasks, so this claim requires either a within-method comparison or citations of prior work showing flows outperform diffusion. Without such ablations or supporting literature, the statement remains speculative.

- No human evaluation, which is necessary and standard in generative modeling papers to support claims of improved perceptual quality.

- Unsubstantiated efficiency claims versus diffusion. The paper states diffusion is slow and flows are efficient, but provides no runtime, sampling-step, or wall-clock comparisons, and no ablation on generation steps, which are essential to support this claim.

- Missing core citations for normalizing flows and flow matching as well as for audio-to-video genreation. The references omit standard normalizing-flow (e.g., NICE/RealNVP/Glow/survey) and flow-matching sources, despite these being central to the method. It would also be benefitial to consider how you work relates to a similar task of audio-to-video (A2V) generation. This area faces the same challenges with alignment and generalization, as seen in "Tango: Text-driven Audio-to-Video Generation," "Diverse and aligned audio-to-video generation via text-to-video model adaptation," and "audio-sync video generation with multi-stream temporal control" and others.

- No figures to explain the method. There are no diagrams of the architecture, masking pipeline, or flow conditioning; this makes it harder to understand the design choices (e.g., selective frame conditioning, iterative flow refinement).

**Questions:**

- Can you provide qualitative examples, such as audio files, since the project site was inaccessible?
- Can you provide an architectural figure to illustrate the MMA/MFG pipeline and the conditioning mechanism?
- Can you clarify the specific flow-based method used?
- Can you provide quantitative data, like runtime, NFE, or wall-clock comparisons, to support your claims of better efficiency over diffusion models?
- Can you provide a direct ablation study comparing the quality of your flow-based generator (MFG) against a similar diffusion-based generator to support your claims?
- Can you provide a human evaluation, as noted in the weaknesses section?

---

### Note · Authors · 2025-11-26

I have read and agree with the venue's withdrawal policy on behalf of myself and my co-authors.